# Towards Explaining Hyperparameter Optimization via Partial Dependence Plots

**Julia Moosbauer**[*,1]                          JULIA.MOOSBAUER@STAT.UNI-MUENCHEN.DA
**Julia Herbinger**[*,1]                           JULIA.HERBINGER@STAT.UNI-MUENCHEN.DE
**Giuseppe Casalicchio**[1]             GIUSEPPE.CASALICCHIO@STAT.UNI-MUENCHEN.DE
**Marius Lindauer**[2]                            LINDAUER@TNT.UNI-HANNOVER.DE
**Bernd Bischl**[1]                               BERND.BISCHL@STAT.UNI-MUENCHEN.DE

[1] *Department of Statistics, Ludwig-Maximilians-University Munich, Munich, Germany*
[2] *Institute of Information Processing, Leibniz University Hanover, Hanover, Germany*

## Abstract

Automated hyperparameter optimization (HPO) approaches often do not provide valuable insights into the effects of different hyperparameters on the final model performance. This lack makes it difficult to trust and understand the automated HPO process and its results. We suggest using interpretable machine learning (IML) to gain insights from the experimental data obtained during HPO and discuss the popular case of Bayesian optimization (BO). BO tends to focus on promising regions with potential high-performance configurations and thus induces a sampling bias. Hence, many IML techniques, like *Partial Dependence Plots* (PDP), carry the risk of generating biased interpretations. By leveraging the posterior uncertainty of the BO surrogate model, we introduce a variant of the PDP with estimated confidence bands. In addition, we propose to partition the hyperparameter space to obtain more confident and reliable PDPs in relevant sub-regions. In an experimental study, we provide quantitative evidence for the increased quality of the PDPs within sub-regions.

## 1. Introduction

AutoML systems mainly return well-performing configurations and leave users without insights into decisions of the optimization process. Many data scientists do not trust the outcome of an AutoML system because of the lack of transparency (Drozdal et al., 2020). Providing insights into the search process may help increase trust and facilitate interactive and exploratory processes: A data scientist could monitor the AutoML process and make changes to it (e.g., restricting or expanding the search space) already *during* optimization to anticipate unintended results. Transparency, trust, and understanding of the inner workings of an AutoML system can be increased by interpreting the internal surrogate model of an AutoML approach. For example, BO trains a surrogate model to approximate the relationship between hyperparameter configurations and model performance. Hence, surrogate models implicitly contain information about the influence of hyperparameters. If the interpretation of the surrogate matches with a data scientist's expectation, confidence in the correct functioning of the system may be increased. If these do not match, it provides an opportunity to look either for bugs in the code or for new theoretical insights.

We propose to analyze surrogate models with methods from IML to provide insights about the results of HPO. We focus on the PDP (Friedman, 2001) as it is a widely used method to visualize the average marginal effect of single features on a black-box model's

prediction. When applied to surrogate models, they provide information on how a specific hyperparameter influences the estimated model performance. However, applying PDPs out of the box to surrogate models might lead to misleading conclusions. Efficient optimizers tend to focus on exploiting promising regions of the hyperparameter space while other regions are less explored, and therefore, a sampling bias in decision space is introduced. Consequently, the surrogate model trained on such experimental data might be biased and thus poorly fit underexplored regions. It follows that PDPs for those surrogate models can be biased when calculated on the entire hyperparameter space.

**Contributions:** We study the problem of sampling bias in experimental data produced by AutoML systems and the resulting bias of the surrogate model and assess its implications on PDPs. We then derive an uncertainty measure for PDPs of probabilistic surrogate models and visualize it as a confidence band around the PDP mean estimate. In addition, we propose a method that splits the hyperparameter space into interpretable sub-regions with varying uncertainties. Thereby, we obtain sub-regions with more reliable and confident estimates for PDPs. In the context of BO, we provide evidence for the usefulness of our proposed methods in an experimental study in which we optimize the architecture and hyperparameters of a deep neural network.

## 2. Background and Related Work

Recent research in AutoML started to question that their evaluation is often solely based on models' predictive performance without considering interpretability (Hutter et al., 2014; Pfisterer et al., 2019; Freitas, 2019; Xanthopoulos et al., 2020). Interpreting AutoML systems can be categorized into interpreting the resulting ML model on the underlying dataset, and interpreting the HPO process itself. In this paper, we focus on the latter aspect.

Let $c : \Lambda \to \mathbb{R}$ be a *black-box* cost function that maps a hyperparameter configuration $\boldsymbol{\lambda} = (\lambda_1, ..., \lambda_d)$ to the model error obtained by a learning algorithm with configuration $\boldsymbol{\lambda}$. The goal of HPO is to find $\boldsymbol{\lambda}^* \in \arg\min_{\boldsymbol{\lambda} \in \Lambda} c(\boldsymbol{\lambda})$. Throughout the paper, we assume that a (probabilistic) surrogate model $\hat{c} : \Lambda \to \mathbb{R}$ is given as an approximation to $c$.

**Partial Dependence for Hyperparameters.** Let $S \subset \{1, 2, ..., d\}$ denote an index set of features, and let $C = \{1, 2, ..., d\} \setminus S$ be its complement. The PDP (Friedman, 2001) of $\hat{c} : \Lambda \to \mathbb{R}$ for a sample $\left(\boldsymbol{\lambda}_C^{(i)}\right)_{i=1,...,n} \sim \mathbb{P}(\boldsymbol{\lambda}_C)$ and hyperparameter(s) $S$ is defined as

$$\hat{c}_S\left(\boldsymbol{\lambda}_S\right) \;=\; \frac{1}{n} \sum\nolimits_{i=1}^{n} \hat{m}\left(\boldsymbol{\lambda}_S, \boldsymbol{\lambda}_C^{(i)}\right), \tag{1}$$

with $\hat{m} : \Lambda \to \mathbb{R}$ denoting the posterior mean. For a fixed $i$, $\hat{m}\left(\boldsymbol{\lambda}_S, \boldsymbol{\lambda}_C^{(i)}\right) : \Lambda_S \to \mathbb{R}$ is called the $i$-th *individual conditional expectation* (ICE) curve (Goldstein et al., 2015). When analyzing the PDP of hyperparameters, we are usually interested in how their values $\boldsymbol{\lambda}_S$ impact model performance uniformly across the hyperparameter space. Therefore, we assume $\mathbb{P}$ to be the uniform distribution over $\Lambda_C$.[1]

**Uncertainty Quantification in PDPs.** Uncertainty of PDPs provides additional information about the reliability of the mean estimator. Hutter et al. (2014) quantify the model uncertainty specifically for random forests as surrogate models in BO by calculating

---

1. This assumption is in line with other works in this domain, like for example (Hutter et al., 2014).

the standard deviation of the marginal predictions of the individual trees. Cafri and Bailey (2016) suggest a bootstrap approach for tree ensembles to quantify the uncertainties of effects based on PDPs. Greenwell (2017) implemented a method that marginalizes over the mean ± standard deviation of the ICE curves for each grid point. A model-agnostic estimate based on uncertainty estimates for probabilistic models has not been proposed so far.

**Subgroup PDPs.** Recently, some researchers started to focus on finding more reliable PDP estimates within subgroups of observations. Molnar et al. (2020), for example, focus on problems in PDP estimation when features are correlated. Grömping (2020) looks at the same problem and also uses subgroup PDPs, where ICE curves are grouped regarding a correlated feature. Britton (2019) applied a clustering approach to group ICE curves to find interactions between features. However, none of the approaches aim at finding subgroups where reliable PDP estimates have low uncertainty. Also to the best of our knowledge, nothing similar exists for analyzing experimental data created by HPO algorithms.

## 3. Biased Sampling in HPO

PDPs for the marginal effect of hyperparameters of surrogate models can be misleading. Here, we show that this problem is due to the sampling and the resulting model bias.

Efficient optimizers tend to exploit promising regions of the hyperparameter space while other regions are less explored. Consequently, predictions of surrogate models are more accurate with less uncertainty in well-explored regions and less accurate with high uncertainty in under-explored regions. This model bias affects the PD estimate (see Figure 1). ICE curves may be biased and less confident if they are computed in poorly learned regions. Assuming uniformly distributed hyperparameters, poorly learned regions are incorporated in the PD estimate with the same weight as well-learned regions. ICE curves belonging to regions with high

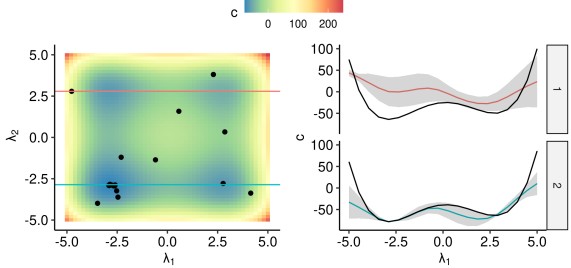

Figure 1: Two ICE curves $\hat{m}\left(\lambda_1, \lambda_2^{(i)}\right)$, $i = 1, 2$, in input space (left), their mean prediction and uncertainty band $\hat{m}\left(\lambda_1, \lambda_2^{(i)}\right) \pm \hat{s}^2\left(\lambda_1, \lambda_2^{(i)}\right)$ against $\lambda_1$ (right) for the surrogate model $\hat{c}$ trained on data created by BO on the $2D$ Styblinski-Tang function.

uncertainty may obfuscate well-learned effects of ICE curves belonging to other regions when they are aggregated to a PDP. Hence, the model bias may also lead to a less reliable PD estimate. PDPs visualizing only the mean estimator of Eq. (1) do not provide insights into the reliability of the PD estimate and how it is affected by the described model bias.

## 4. Quantifying Uncertainty in PDPs

Pointwise uncertainty estimates of a probabilistic model provide insights into the reliability of the prediction $\hat{c}(\boldsymbol{\lambda})$ for a specific configuration $\boldsymbol{\lambda}$. This uncertainty directly correlates with how explored the region around $\boldsymbol{\lambda}$ is. Hence, including the model's uncertainty structure into the PD estimate enables a user to understand in which regions the PDP is more reliable and which parts of the PDP have to be interpreted with caution. We now extend the PDP

of Eq. (1) to probabilistic surrogate models $\hat{c}$ (e.g., a GP). Let $\boldsymbol{\lambda}_S$ be a fixed grid point and $\left(\boldsymbol{\lambda}_C^{(i)}\right)_{i=1,\dots,n} \sim \mathbb{P}\left(\boldsymbol{\lambda}_C\right)$. The vector of predicted performances at the grid point $\boldsymbol{\lambda}_S$ is $\hat{\boldsymbol{c}}\left(\boldsymbol{\lambda}_S\right) = \left(\hat{c}\left(\boldsymbol{\lambda}_S, \boldsymbol{\lambda}_C^{(i)}\right)\right)_{i=1,\dots,n}$ with (posterior) mean $\hat{\boldsymbol{m}}\left(\boldsymbol{\lambda}_S\right) := \left(\hat{m}\left(\boldsymbol{\lambda}_S, \boldsymbol{\lambda}_C^{(i)}\right)\right)_{i=1,\dots,n}$ and a (posterior) covariance $\hat{\boldsymbol{K}}\left(\boldsymbol{\lambda}_S\right) := \left(\hat{k}\left(\left(\boldsymbol{\lambda}_S, \boldsymbol{\lambda}_C^{(i)}\right), \left(\boldsymbol{\lambda}_S, \boldsymbol{\lambda}_C^{(j)}\right)\right)\right)_{i,j=1,\dots,n}$. Thus, $\hat{c}_S\left(\boldsymbol{\lambda}_S\right) = \frac{1}{n}\sum_{i=1}^n \hat{c}\left(\boldsymbol{\lambda}_S, \boldsymbol{\lambda}_C^{(i)}\right)$ is a random variable itself. The variance of $\hat{c}_S\left(\boldsymbol{\lambda}_S\right)$ is

$$\hat{s}_S^2(\boldsymbol{\lambda}_S) \quad = \quad \mathbb{V}_{\hat{\boldsymbol{c}}}\left[\hat{c}_S\left(\boldsymbol{\lambda}_S\right)\right] = \mathbb{V}_{\hat{\boldsymbol{c}}}\left[\frac{1}{n}\sum_{i=1}^n \hat{c}\left(\boldsymbol{\lambda}_S, \boldsymbol{\lambda}_C^{(i)}\right)\right] = \frac{1}{n^2}\mathbf{1}^\top \hat{\boldsymbol{K}}\left(\boldsymbol{\lambda}_S\right)\mathbf{1}. \tag{2}$$

It is important to ensure that the kernel is correctly specified. Eq. (2) can be approximated empirically by treating the pairwise covariances as unknown, i.e.:

$$\hat{s}_S^2\left(\boldsymbol{\lambda}_S\right) \quad \approx \quad \frac{1}{n}\sum_{i=1}^n \hat{\boldsymbol{K}}\left(\boldsymbol{\lambda}_S\right)_{i,i}. \tag{3}$$

In Appendix A.2, we show that this is less sensitive to misspecifications in the kernel. Note that the variance estimate and the mean estimate can also be applied to other probabilistic models, such as GAMLSS, transformation trees, or a random forest.

## 5. Partial Dependence Plots on Sub-regions

As discussed in Section 3, (efficient) optimization may imply that the design is biased, which in turn can produce misleading analyses when IML methods are naively applied. We now aim to identify sub-regions $\Lambda' \subset \Lambda$ of the hyperparameter space in which the PDP can be estimated with high confidence, and separate those from sub-regions in which it cannot be estimated reliably. We identify sub-regions in which poorly-learned effects do not obfuscate the well-learned effects along each grid point, allowing the user to draw conclusions more confidently in these locations. By splitting the entire hyperparameter space in disjoint and interpretable sub-regions which can be visualized and interpreted individually but also aggregated to a global view, the user gets a better understanding of how the optimazation and exploration of the sampling process influences hyperparameter effects. Thus, to combine these requirements, we introduce a simple, but interpretable tree-based partitioning procedure which splits the entire hyperparameter space in disjoint and interpretable sub-regions in such a way that we receive more confident PDP estimates for well-explored regions and less reliable estimates in under-explored regions. This procedure can also be visualized via a tree structure as shown in Figure 2 for one partitioning step.

The PD estimate on the *entire* hyperparameter space $\Lambda$ is computed by drawing the sample used for the Monte Carlo estimate $(\boldsymbol{\lambda}_C^{(i)})_{i\in\mathcal{N}} \sim \mathbb{P}(\boldsymbol{\lambda}_C)$, $\mathcal{N} := \{1, 2, \dots, n\}$. The PD estimate on a *sub-region* $\Lambda' \subset \Lambda$ will be approximated w.r.t. $(\boldsymbol{\lambda}_C^{(i)})_{i\in\mathcal{N}'}$ with $\mathcal{N}' = \{i \in \mathcal{N}\}_{\boldsymbol{\lambda}^{(i)}\in\Lambda'}$ only. Since we are interested in the marginal effect of the hyperparameter(s) $S$ at each $\boldsymbol{\lambda}_S \in \Lambda_S$ we will visualize the PD for the whole range $\Lambda_S$. Thus, we want to have sub-regions of the form $\Lambda' = \Lambda_S \times \Lambda_C'$ with $\Lambda_C' \subset \Lambda_C$. This corresponds to the average over the ICE curves $i \in \mathcal{N}'$. The criterion to evaluate a specific partitioning is based on the idea of grouping ICE curves with similar uncertainty structure. We evaluate

the impurity of a PD estimate on a sub-region $\Lambda'$ by the associated set of observations $\mathcal{N}' = \{i \in \mathcal{N}\}_{\boldsymbol{\lambda}_C^{(i)} \in \Lambda'_C}$, also referred to as nodes, as follows: For each grid point $\boldsymbol{\lambda}_S$, we use the L2 loss in $L\left(\boldsymbol{\lambda}_S, \mathcal{N}'\right)$ to evaluate how uncertainty varies across all ICE estimates $i \in \mathcal{N}'$ using $\hat{s}_{S|\mathcal{N}'}^2\left(\boldsymbol{\lambda}_S\right) := \frac{1}{|\mathcal{N}'|}\sum_{i \in \mathcal{N}'}\hat{s}^2\left(\boldsymbol{\lambda}_S, \boldsymbol{\lambda}_C^{(i)}\right)$ and aggregate over all grid points in $\mathcal{R}_{L2}(\mathcal{N}')$:

$$L\left(\boldsymbol{\lambda}_S, \mathcal{N}'\right) = \sum_{i \in \mathcal{N}}\left(\hat{s}^2\left(\boldsymbol{\lambda}_S, \boldsymbol{\lambda}_C^{(i)}\right) - \hat{s}_{S|\mathcal{N}'}^2\left(\boldsymbol{\lambda}_S\right)\right)^2 \text{ and } \mathcal{R}_{L2}(\mathcal{N}') = \sum_{g=1}^{G}L(\boldsymbol{\lambda}_S^{(g)}, \mathcal{N}'). \quad (4)$$

The partitioning based on CART (Breiman et al., 1984) is described in Appendix B.

The chosen impurity measure of Eq. (4) measures the pointwise $L_2$-distance between ICE curves of the variance function $\hat{s}^2(\boldsymbol{\lambda}_S, \boldsymbol{\lambda}_C^{(i)})$ and its PD estimate $\hat{s}_{S|\mathcal{N}'}^2\left(\boldsymbol{\lambda}_S\right)$ within a sub-region $\mathcal{N}'$. It follows that the splits are based on ICE curve similarities regarding the uncertainty measure $\hat{s}_S^2$. This seems reasonable, as ICE curves in well-explored regions of the search space should, on average, have a lower uncertainty than the ones in less-explored regions. In particular, we expect the PDP estimate in the sub-regions associated with low costs $c$ (and thus high relevance for a user) to be more confident in well-explored re-

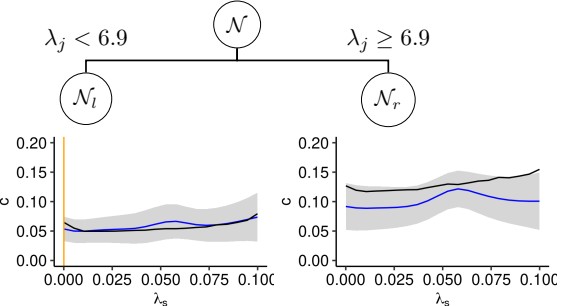

Figure 2: The figure shows two estimated PDPs (blue) and 95% confidence bands after one partitioning step. The orange vertical line marks the value of hyperparameter $S$ of the optimal configuration, the black curve shows the true PD estimate.

gions of $\boldsymbol{\lambda}_S$. This is illustrated in Figure 2 where the confidence band in the left sub-region decreased compared to the confidence band of the global PDP especially for grid points close to the optimal configuration of $\boldsymbol{\lambda}_S$. We argue that ICE curves of the variance function are on average lower and show a similar trend for well-explored regions of the $\Lambda$. Thus, by splitting according to curve similarities, we receive at least one interpretable sub-region (containing the optimal configuration) where the PDP can be interpreted more reliably.

## 6. Experimental Analysis

In this section, we show that the proposed tree-based partitioning procedure allows us to identify an interpretable sub-region that yields a more reliable and confident PDP estimate.

We investigate HPO in the context of a surrogate benchmark (Eggensperger et al., 2015) based on the LCBench data (Zimmer et al., 2021), incl. 2000 randomly sampled configurations of the small Auto-PyTorch space on 35 datasets. For each dataset, we trained a random forest as empirical performance model which predicts the balanced validation error achieved by Auto-Pytorch Tabular for a given configuration. These surrogates serve as objective that is optimized via BO. By design, the objective is cheaper to evaluate and allows to infer a ground-truth. For each task, we ran BO to obtain the optimal hyperparameters and computed PDPs on the final surrogate model (cf. Appendix C.1 for more details).

We measure the reliability of a PDP, i.e., the degree to which a user can rely on the estimate of the PD estimate, by comparing it to the true PD $c_S(\boldsymbol{\lambda}_S)$ computed on the true

objective function $c$. More specifically, for every grid point $\boldsymbol{\lambda}_S^{(g)}$, we compute the negative log-likelihood (NLL) of $c_S(\boldsymbol{\lambda}_S)$ under the distribution of $\hat{c}_S(\boldsymbol{\lambda}_S)$ pointwise for every grid point $\boldsymbol{\lambda}_S^{(g)}$. We measure the confidence by assessing $\hat{s}_S(\boldsymbol{\lambda}_S)$ pointwise for every grid point. We consider the mean confidence (MC) across all grid points $\frac{1}{G}\sum_{g=1}^{G}\hat{s}\left(\boldsymbol{\lambda}_S^{(g)}\right)$ as well as the confidence at the grid point closest to $\hat{\boldsymbol{\lambda}}_S$ (OC), with $\hat{\boldsymbol{\lambda}}$ being the best configuration evaluated by the optimizer. When evaluating the performance of the tree-based partitioning, we report the above metrics on *that* sub-region which contains the best configuration evaluated by the optimizer, assuming that this region is of particular interest for a user of HPO. We compared the PDP in sub-regions after 6 splits with the global PDP by computing the relative improvement in confidence (MC and OC) and NLL of the sub-regional PDP compared to those metrics for the global PDP. The MC of the PDPs is on average reduced by 30% to 52%, depending on the hyperparameter (cf. Table 1). At the optimal configuration $\hat{\boldsymbol{\lambda}}_S$ the improvement even increases by $50\% - 62\%$. PDP estimates for all hyperparameters are on average clearly more confident in the relevant sub-regions when compared to the global ones, especially in the region around the optimal configuration $\hat{\boldsymbol{\lambda}}_S$. Furthermore, NLL even improves while the MC decreases. A more detailed overview on dataset level and a benchmark with other impurity measures can be found in Appendix C.2.

## 7. Conclusion

We showed that PD estimates for surrogate models fitted on experimental data generated by HPO algorithms can be unreliable due to an underlying sampling bias. We extended PDPs by an uncertainty estimate to provide the user with more information regarding the reliability of the mean estimator. We introduced a tree-based partitioning approach for PDPs where we leverage the uncertainty estimator to decompose the hyperparameter space into interpretable, disjoint sub-regions. Our experiments show that we generate, on average, more confident and reliable PD estimates in the sub-region containing the

| Hyperparameter | $\delta$ MC | $\delta$ OC | $\delta$ NLL |
|---|---|---|---|
| Batch size | 41 (15) | 62 (14) | 20 (20) |
| Learning rate | 50 (14) | 58 (14) | 18 (21) |
| Max. dropout | 50 (15) | 62 (12) | 17 (18) |
| Max. units | 51 (15) | 59 (13) | 25 (22) |
| Momentum | 52 (15) | 58 (13) | 20 (22) |
| Number of layers | 31 (16) | 51 (17) | 14 (33) |
| Weight decay | 36 (23) | 61 (13) | 12 (20) |

Table 1: Rel. improvement of MC, OC and NLL on hyperparameter level in %. The table shows the respective mean (standard deviation) of the average relative improvement over 30 replications for each dataset and 6 splits.

optimal configuration compared. While we mainly discussed GP surrogates on a numerical hyperparameter space, our methods are applicable to all kinds of distributional regression models and for mixed hyperparameter spaces. In future, we will study our method on more complex, hierarchical configuration spaces for neural architecture search.

## Acknowledgements

This work has been partially supported by the German Federal Ministry of Education and Research (BMBF) under Grant No. 01IS18036A. The authors of this work take full responsibilities for its content.

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

## Appendix A. Uncertainty Estimation

### A.1 Choice of Uncertainty Quantification

Besides using the uncertainty estimate of the surrogate model to quantify the uncertainty for the PDP mean estimate (our method), it is also possible to estimate uncertainty w.r.t. the variance over different ICE curves (Greenwell, 2017). However, if the uncertainty was estimated via computing the variance over ICE curves, we describe how the *levels of the mean prediction vary* along the $\boldsymbol{\lambda}_S$ dimensions. In contrast, we propose to capture *model uncertainty* along the $\boldsymbol{\lambda}_S$ dimensions. For example, consider a constant surrogate function $\hat{c}(\boldsymbol{\lambda}) = \gamma$ with high uncertainty estimation $\hat{s}^2(\boldsymbol{\lambda}) = 100$. Computing the variance over ICE curves on this example will result in an uncertainty estimate of 0 (all ICE curves are identical). Our method, however, would return a variance estimate of 100 and thus capture model uncertainty.

### A.2 Covariance Estimates under Misspecification of Kernels

In order to provide evidence for the claim that estimate (2) is more sensitive to misspecifications in the kernel (and thus in the covariance structure) than (3), we performed some prior experiments. We assume that we are given an objective function that is generated by a Gaussian process (GP) with a Matérn-3/2 kernel. In our experiments, that function was created by fitting a GP on tuples $\left(\boldsymbol{\lambda}^{(i)}, y^{(i)}\right)_{i=1,\dots,30}$, with $\boldsymbol{\lambda}^{(i)} \sim \text{Unif}\left([-5,5]^d\right)$ and $y^{(i)}$ corresponding to the value of the $d$-dimensional Styblinski Tang function for $\boldsymbol{\lambda}^{(i)}$. The posterior mean of this GP will further serve as our true objective $c$ to pretend that we know the correct kernel specification of the ground-truth.

Subsequently, we fit both a GP surrogate model with correctly specified kernel (i.e., a Matérn-3/2 kernel) and a surrogate model with a misspecified kernel (in our case, we chose a Gaussian kernel) to the data $\left(\boldsymbol{\lambda}^{(i)}, c\left(\boldsymbol{\lambda}^{(i)}\right)\right)_{i=1,\dots,30}$. In both cases, we compute the partial dependence plots (PDPs) for $\boldsymbol{\lambda}_1$ with both variance estimates (2) and (3) and measure the negative log-likelihood of $c_S$ under the respective estimated PDP. We performed 50 repetitions of the experiments for $d \in \{3, 5, 8\}$, respectively.

Figure 3 shows that the median of the NLL across all 50 replications is *slightly* lower for the covariance estimate (2). However, the variance of the NLL is much higher for estimate (2) as compared to (3). Table 2 confirms that, when using variance estimate (2), the standard deviation of the NLL values is lower. We conclude that the reliability of the estimate is particularly sensitive to a correct choice of the kernel function. The NLL for the PDPs computed with variance estimate (3) is - independent of whether the kernel is correctly specified or not - less sensitive to misspecifications in the kernel.

## Appendix B. Tree-based Partitioning Algorithm

The steps to partition the hyperparameter space $\Lambda$ into two sub-regions based on an i.i.d. uniformly distributed dataset $(\boldsymbol{\lambda}_C^{(i)})_{i \in \mathcal{N}} \in \Lambda_C$, $\mathcal{N} = \{1, \dots, n\}$, are shown in Algo. 1. The procedure is based on the CART splitting algorithm introduced by Breiman et al. (1984). It is recursively repeated until a user-defined stopping criterion is met (e.g., a maximum number of splits, a minimum size of a region, or a minimum improvement in each node). For

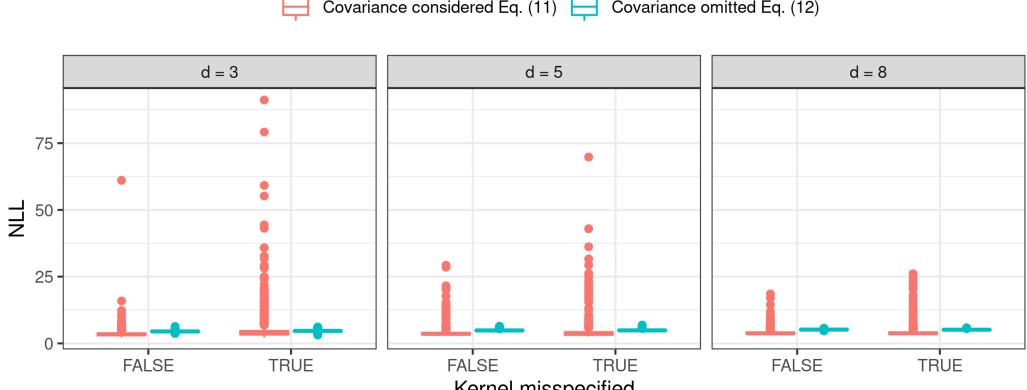

Figure 3: The figures show the Negative Log-Likelihood (NLL) of the true PDP $c_1(\boldsymbol{\lambda}_1)$ under the estimated PDPs with variance estimates (2) and (3) and for a misspecified kernel (Gaussian) and a correctly specified kernel (Matérn-3/2), respectively.

| | Correct specification | | Misspecification | |
| d | Estimate (3) | Estimate (2) | Estimate (3) | Estimate (2) |
|---|---|---|---|---|
| 3 | 3.61 (2.02) | 4.47 (0.27) | 5.10 (5.91) | 4.62 (0.32) |
| 5 | 3.93 (2.00) | 4.87 (0.23) | 4.33 (3.72) | 4.89 (0.28) |
| 8 | 4.05 (1.12) | 5.18 (0.14) | 4.24 (2.12) | 5.13 (0.17) |

Table 2: The table shows the Negative Log-Likelihood (NLL) of the true PDP $c_1(\boldsymbol{\lambda}_1)$ under the estimated PDPs with variance estimates (2) and (3) and for a misspecified kernel (Gaussian) and a correctly specified kernel (Matérn-3/2), respectively. Shown are the mean across 50 replications, and the standard deviation in brackets.

the leaf nodes, we calculate PDP and uncertainty estimate within the regarded sub-region by aggregating the respective ICE curves belonging to those nodes. The identified sub-regions belonging to the leaf nodes can be deduced from split variables and split points in every step. To receive a more confident PDP mean estimate, we would like to group those ICE curves that represent the same ranges of $\boldsymbol{\lambda}_S$ well by exhibiting low and similar uncertainty in this range. This is shown in Figure 4, where samples with high uncertainty over the entire range of $\boldsymbol{\lambda}_S$ are grouped together (right sub-region). Samples with low uncertainty close to the optimal configuration of $\boldsymbol{\lambda}_S$ and increasing uncertainties for less suitable configurations are grouped together by curve similarities in the left sub-region.

**Number of Splits** Which PDPs are most interesting to look at depends on the question the user would like to answer. If the main interest lies in understanding the optimization and exploring the sampling process, a user might want to keep the number of sub-regions relatively low by performing only a few partitioning steps. Subsequently, one would investigate the overall structure of the sub-regions and the individual sub-regional PDPs. If the user is more interested in interpreting hyperparameter effects only in the most relevant sub-region(s) around the best configurations evaluated by the optimizer, a user potentially may want to split deeper and only look at sub-regions that are more confident than the global PDP.

**Algorithm 1:** Tree-based Partitioning

$\quad$ **input:** $\mathcal{N}$
$\quad$ **output:** $\mathcal{N}_l^{t^*}$ and $\mathcal{N}_r^{t^*}$
$\quad$ **for** $j \in C$ **do**
$\quad\quad$ **for** Every split $t$ on hyperparameter $\lambda_j$ **do**
$\quad\quad\quad$ $\mathcal{N}_l^t = \{i \in \mathcal{N}\}_{\lambda_j^{(i)} \leq t}$
$\quad\quad\quad$ $\mathcal{N}_r^t = \{i \in \mathcal{N}\}_{\lambda_j^{(i)} > t}$
$\quad\quad\quad$ $\mathcal{I}(t) = \mathcal{R}_{L_2}(\mathcal{N}_l^t) + \mathcal{R}_{L_2}(\mathcal{N}_r^t)$
$\quad\quad$ **end for**
$\quad\quad$ Choose $t_{\lambda_j}^* \in \text{arg min}_t \, \mathcal{I}(t)$
$\quad$ **end for**

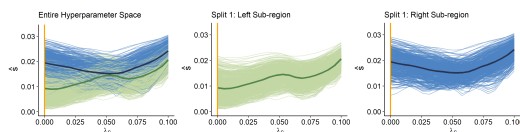

Figure 4: The figure shows ICE curves of $\hat{s}$ of $\boldsymbol{\lambda}_S$. Left: Defined groups in first split with the darker lines representing the PDPs within the sub-region; middle (right): ICE curves after first split in left (right) sub-region. The orange vertical line marks the value $\lambda_S$ of the optimal configuration.

## Appendix C. Experimental Analysis

### C.1 Experimental Design

All experiments only require CPUs (and no GPUs) and were computed on a Linux cluster with 28-way Haswell-EP nodes, 1 core per node, and a memory limit of 2.2 GB per node.

The computational complexity of the PDP estimation with uncertainty is $\mathcal{O}(G \cdot n) \cdot \mathcal{O}(\hat{c})$, with $\mathcal{O}(\hat{c})$ being the runtime complexity of single surrogate prediction, $n$ denoting the size of the dataset to compute the Monte Carlo estimate and $G$ being the number of grid points. In the context of HPO, the general assumption is that evaluation time of $\hat{c}$ is negligibly low as compared to evaluation $c$. So we argue that the runtime complexity of computing a PDP with uncertainty estimate can be neglected in this context. When computing ICE curves and their variance estimates beforehand, the algorithmic complexity of Algorithm 1 corresponds to the algorithmic complexity of the tree splitting (Breiman et al., 1984).

In our experiments, the runtimes to compute the PDPs and perform the tree splitting lies within a few minutes. We consider them to be negligible and will thus not report these.

All experimental data was downloaded from the LCBench project[2]. As empirical performance model, we fitted a random forest (ranger) to approximate the relationship between hyperparameters and balanced error rate (BER). For every dataset, we performed a random search with 500 iterations and evaluation via 3-fold cross validation to choose reasonable for the hyperparameters represented in Table 3. The empirical performance model acts as ground-truth in our experiments, and thus, we denote it by $c$. This function was used to compute the true PDP $c_S$.

We computed an initial random design of size $2 \cdot d^3$. We performed BO with a GP surrogate model with a Matérn-3/2 kernel and the LCB acquisition function $a(\boldsymbol{\lambda}) = \hat{m}(\boldsymbol{\lambda}) + \tau \cdot \hat{s}(\boldsymbol{\lambda})$ with different values $\tau \in \{1\}$. A nugget effect was modeled. The maximum budget per BO run was set to 200 objective function evaluations. We denote the best evaluated configuration, measured by $\hat{c}$, by $\hat{\boldsymbol{\lambda}}$.

---

2. https://github.com/automl/LCBench
3. The initial design was fixed across replications

| Name | Range | log | type |
|---|---|---|---|
| # trees | $[10, 500]$ | yes | int |
| mtry | {true, false} | no | bool |
| Min. node size | $[1, 5]$ | no | int |
| # random splits | $[1, 100]$ | no | int |

Table 3: Hyperparameter space of the random forest that was tuned over to compute the empirical performance model.

| Name | Range | log | type |
|---|---|---|---|
| Number of layers | $[1, 5]$ | no | int |
| Max. number of units | $[64, 512]$ | yes | int |
| Batch size | $[16, 512]$ | yes | int |
| Learning rate (SGD) | $[1e^{-4}, 1e^{-1}]$ | yes | float |
| Weight decay | $[1e^{-5}, 1e^{-1}]$ | no | float |
| Momentum | $[0.1, 0.99]$ | no | float |
| Max. dropout rate | $[0.0, 1.0]$ | no | float |

Table 4: Hyperparameter space 1 of Auto-PyTorch Tabular.

Based on the last surrogate model, we performed the partitioning in Algorithm 1 for a total number of 6 splits, with the different splitting criteria (see Section C.2), with PDPs being computed with a $G = 20$ equidistant grid points, and $n = 1000$ samples for the Monte Carlo approximation[4].

All PDPs are computed with regards to a single feature, for a grid of $G = 20$ equidistant points, and the Monte Carlo estimate is computed with $n = 1000$ samples.

We illustrate the confidence of a PDP by the width of its confidence bands $\hat{m}_S(\boldsymbol{\lambda}_S) \pm q_{1-\alpha/2} \cdot \hat{s}_S(\boldsymbol{\lambda}_S)$, with $q_{1-\alpha/2}$ denoting the $(1-\alpha/2)$-quantile of a standard normal distribution.

## C.2 Detailed Results

In Section 6 we evaluated the reliability of PDP estimation for the partitioning procedure proposed in Section 5. The results presented in Section 5 are aggregated over a total number of 35 different datasets. In Tables 5 and 6 the relative improvement of the mean confidence (MC) and negative loglikelihood (NLL) are presented on dataset level. The mean and standard deviation are averaged over all hyperparameters. Furthermore, the mean values of the features providing the highest and lowest relative improvement for each dataset are reported. Following on that, Table 7 shows for each hyperparameter the the number of datasets for which the respective hyperparameter led to the highest (lowest) relative improvement for both evaluation metrics.

To further study our suggested method, we now highlight a few individual experiments. We chose one iteration of the *shuttle* dataset. In the upper panel of Figure 5, we see that the true PDP estimate for *max. number of units* is decreasing, while the globally estimated PDP trend is increasing, and thus misleading. Although the confidence band already signals that the PDP cannot be reliably interpreted on the entire hyperparameter space, it remains challenging to draw any conclusions from it.

| | MC | | NLL | |
|---|---|---|---|---|
| Hyperparameter | # $\mu_h$ | # $\mu_l$ | # $\mu_h$ | # $\mu_l$ |
| Batch size | 1 | 3 | 3 | 4 |
| Learning rate | 6 | 2 | 6 | 3 |
| Max. dropout | 9 | 1 | 2 | 1 |
| Max. units | 4 | | 7 | |
| Momentum | 8 | | 7 | 3 |
| Number of layers | 3 | 14 | 9 | 11 |
| Weight decay | 4 | 15 | 1 | 13 |

Table 7: Number of datasets each of the hyperparameters had the highest $\mu_h$ and lowest $\mu_l$ average relative improvement w.r.t. MC.

After performing 6 splits, we receive a confident and reliable PD estimate on an interpretable sub-region. The same plots are depicted for the hyperparameter *batch size* in the lower

---

4. The grid and the data used to compute the Monte Carlo estimate was fixed across replications

| Dataset | $\mu$ | $\sigma$ | $\mu_h$ | $\mu_l$ |
|---|---|---|---|---|
| adult | 34 | 6 | 38 | 25 |
| airlines | 49 | 20 | 61 | 3 |
| albert | 57 | 26 | 78 | 14 |
| Amazon_employee_access | 58 | 17 | 69 | 21 |
| APSFailure | 46 | 17 | 60 | 22 |
| Australian | 41 | 7 | 46 | 32 |
| bank-marketing | 29 | 13 | 45 | 15 |
| blood-transfusion-service | 34 | 20 | 39 | 13 |
| car | 44 | 17 | 51 | 32 |
| christine | 47 | 14 | 54 | 19 |
| cnae-9 | 66 | 26 | 83 | 7 |
| connect-4 | 47 | 14 | 56 | 17 |
| covertype | 41 | 17 | 53 | 12 |
| credit-g | 57 | 21 | 69 | 7 |
| dionis | 49 | 21 | 63 | 5 |
| fabert | 64 | 21 | 75 | 18 |
| Fashion-MNIST | 41 | 12 | 47 | 18 |
| helena | 43 | 16 | 52 | 8 |
| higgs | 42 | 14 | 52 | 17 |
| jannis | 35 | 13 | 44 | 19 |
| jasmine | 46 | 11 | 56 | 27 |
| jungle_chess_2pcs_raw | 33 | 15 | 44 | 6 |
| kc1 | 33 | 12 | 41 | 17 |
| KDDCup09_appetency | 52 | 21 | 63 | 3 |
| kr-vs-kp | 46 | 14 | 56 | 26 |
| mfeat-factors | 56 | 16 | 70 | 29 |
| MiniBooNE | 36 | 14 | 42 | 18 |
| nomao | 30 | 6 | 34 | 22 |
| numerai28.6 | 60 | 28 | 76 | -3 |
| phoneme | 29 | 7 | 32 | 25 |
| segment | 53 | 21 | 66 | 10 |
| shuttle | 48 | 11 | 58 | 32 |
| sylvine | 37 | 6 | 42 | 29 |
| vehicle | 34 | 8 | 41 | 30 |
| volkert | 44 | 16 | 55 | 12 |

Table 5: Relative improvement of MC on dataset level. The table shows the mean ($\mu$) and standard deviation ($\sigma$) of the relative improvement (in %) over all 7 hyperparameters and 30 runs after 6 splits. Additionally the mean value of the hyperparameter with the highest ($\mu_h$) and lowest ($\mu_l$) mean improvement are shown.

| Dataset | $\mu$ | $\sigma$ | $\mu_h$ | $\mu_l$ |
|---|---|---|---|---|
| adult | 13 | 6 | 23 | 8 |
| airlines | 17 | 9 | 23 | 1 |
| albert | 31 | 13 | 40 | 6 |
| Amazon_employee_access | -0 | 36 | 29 | -35 |
| APSFailure | 15 | 7 | 23 | 6 |
| Australian | 12 | 14 | 23 | -4 |
| bank-marketing | 7 | 9 | 17 | -1 |
| blood-transfusion-service | 6 | 17 | 10 | -8 |
| car | 26 | 32 | 35 | 10 |
| christine | 10 | 11 | 17 | 1 |
| cnae-9 | 67 | 37 | 93 | -11 |
| connect-4 | -4 | 38 | 22 | -84 |
| covertype | 28 | 13 | 38 | 8 |
| credit-g | 41 | 24 | 81 | 2 |
| dionis | 47 | 55 | 144 | -18 |
| fabert | 37 | 17 | 54 | 8 |
| Fashion-MNIST | 15 | 11 | 28 | 2 |
| helena | -20 | 31 | -9 | -35 |
| higgs | 20 | 12 | 33 | -2 |
| jannis | 17 | 7 | 21 | 8 |
| jasmine | 6 | 14 | 24 | -11 |
| jungle_chess_2pcs_raw | 9 | 15 | 24 | -7 |
| kc1 | 11 | 10 | 17 | 4 |
| KDDCup09_appetency | 23 | 28 | 62 | -33 |
| kr-vs-kp | 9 | 35 | 43 | -17 |
| mfeat-factors | 25 | 19 | 51 | 10 |
| MiniBooNE | 9 | 14 | 17 | -8 |
| nomao | 8 | 6 | 16 | 3 |
| numerai28.6 | 17 | 9 | 23 | 4 |
| phoneme | 11 | 7 | 17 | 5 |
| segment | 22 | 57 | 41 | -31 |
| shuttle | 35 | 24 | 84 | 19 |
| sylvine | 14 | 17 | 38 | -0 |
| vehicle | 0 | 20 | 9 | -14 |
| volkert | 23 | 18 | 40 | 5 |

Table 6: Relative improvement of the NLL on dataset level. The table shows the mean ($\mu$) and standard deviation ($\sigma$) of the relative improvement (in %) over all 7 hyperparameters and 30 runs after 6 splits. Additionally the mean value of the feature with the highest ($\mu_h$) and lowest ($\mu_l$) mean improvement are shown.

panel of Figure 5. This example illustrates that the confidence band might not always shrink uniformly over the entire range of $\boldsymbol{\lambda}_S$ during the partitioning, but often particularly around the optimal configuration $\hat{\boldsymbol{\lambda}}_S$.

**Split Criteria** In Section 5, we introduced Eq. 4 as split criteria within the tree-based partitioning of Algorithm 1. This measure is based on splitting ICE curves based on curve similarities, which is especially suitable in the underlying context as explained in Section 5. However, we also compared it to two other measures that are based on uncertainty estimates provided by the probabilistic surrogate model. The first one is also based on ICE curves of the variance function $\hat{s}^2(\boldsymbol{\lambda}_S, \boldsymbol{\lambda}_C^{(i)})$ and its PD estimate $\hat{s}^2_{S|\mathcal{N}'}(\boldsymbol{\lambda}_S)$ within a sub-region $\mathcal{N}'$. However, instead of minimizing the distance between curves and group the associated

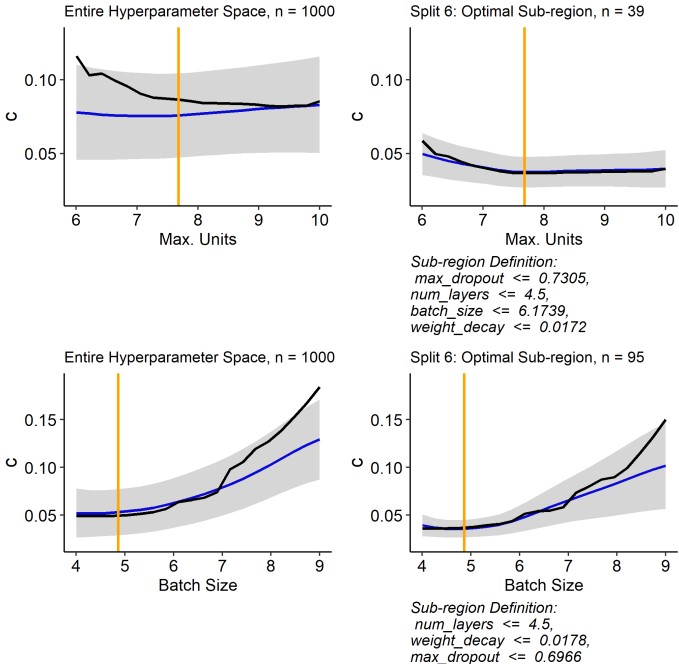

Figure 5: PDP (blue) and confidence band (grey) of the GP for hyperparameter *max. number of units*The black line shows the PDP of the meta surrogate model representing the true PDP estimate. The orange vertical line marks the optimal configuration $\hat{\boldsymbol{\lambda}}_S$. The relative improvements from the global PDP (left) to the sub-regional PDP after 6 splits (right) are : $\delta$ MC $= 61.6\%$ , $\delta$ OC $= 63.5\%$ , $\delta$ NLL $= 48.6\%$ .

ICE curves regarding similar behavior, we can also minimize the area under ICE curves of the variance function. The reasoning for this is as follows: If we aim for tight confidence bands over the entire range of $\Lambda_S$, we want the ICE curves of the variance function to be - on average - as low as possible. This is equivalent to minimizing the average area under ICE curves of the variance function. Thus, the calculation of Eq. 4 changes such that we first calculate the average area between each ICE curve of the uncertainty function and the respective sub-regional PDP

$$L\left(\boldsymbol{\lambda}_S, i\right) = \frac{1}{G} \sum_{g=1}^{G} \left(\hat{s}^2 \left(\boldsymbol{\lambda}_S^{(g)}, \boldsymbol{\lambda}_C^{(i)(g)}\right) - \hat{s}^2_{S|\mathcal{N}'} \left(\boldsymbol{\lambda}_S^{(g)}\right)\right),$$

where $\hat{s}^2_{S|\mathcal{N}'} \left(\boldsymbol{\lambda}_S^{(g)}\right) := \frac{1}{|\mathcal{N}'|} \sum_{i \in \mathcal{N}'} \hat{s}^2 \left(\boldsymbol{\lambda}_S^{(g)}, \boldsymbol{\lambda}_C^{(i)(g)}\right)$ , and aggregate the quadratic value of it over all observations in the respective sub-region:

$$\mathcal{R}_{area}(\mathcal{N}') = \sum_{i \in \mathcal{N}'} L(\boldsymbol{\lambda}_S, i)^2. \tag{5}$$

Second, we also used the uncertainty estimates of the probabilistic surrogate model for each observation of the testdata itself to define an impurity measure. Therefore we calculated the squared deviation of each observation to the mean uncertainty within the respective node. Hence, compared to the other two approaches, we do not group curves but the observations itself regarding their uncertainty. We further refer to this approach as the *variance (var)* approach.

As a third measure that is not based on the uncertainty estimates, we used the MSE of the posteriori mean estimate of the surrogate model as split criterion. This is the most commonly used measure for regression trees and hence a solid baseline measure.

We compared the four impurity measures for the partitioning procedure over all datasets and hyperparameters. We compare the results that we presented in Section 6 with the according results for the other three measures in Table 8. The impurity measure based on curve similarities that we used for our analysis (L2) outperforms the other three measures on average for all hyperparameters regarding MC and especially regarding OC. With regards to NLL there is not one measure which outperforms all others, but rather all measures perform on average over all hyperparameters equally good.

| | $\delta$ MC (in %) | | | | $\delta$ OC (in %) | | | | $\delta$ NLL (in %) | | | |
|---|---|---|---|---|---|---|---|---|---|---|---|---|
| Hyperparameter | L2 | area | var | mean | L2 | area | var | mean | L2 | area | var | mean |
| Batch size | 41 | 40 | 38 | 36 | 62 | 58 | 55 | 53 | 20 | 19 | 16 | 19 |
| Learning rate | 50 | 50 | 50 | 42 | 58 | 57 | 57 | 51 | 18 | 18 | 18 | 19 |
| Max. dropout | 50 | 49 | 47 | 41 | 62 | 61 | 58 | 53 | 17 | 18 | 17 | 15 |
| Max. units | 51 | 51 | 50 | 45 | 59 | 58 | 58 | 53 | 25 | 24 | 25 | 25 |
| Momentum | 52 | 51 | 51 | 43 | 58 | 57 | 57 | 53 | 20 | 20 | 20 | 16 |
| Number of layers | 31 | 30 | 29 | 25 | 51 | 46 | 46 | 45 | 14 | 15 | 15 | 13 |
| Weight decay | 36 | 35 | 34 | 29 | 61 | 53 | 51 | 52 | 12 | 12 | 11 | 10 |

Table 8: Comparison of different impurity measures regarding the relative improvement of MC, OC and NLL on hyperparameter level. The table compares the results of Table 1 (L2) with the according results for the impurity measure based on Eq. 5 (area), the *variance* measure (var) and the *mean* measure.

**Increased confidence with more splits**   Furthermore, it needs to be noted that by using our method the mean confidence and NLL improve on average if we use six splits. However, this does not mean that they improve by design when splitting into sub-regions. As shown in Tables 5 and 6, improvements heavily depend on dataset and HP. Different factors influence the optimal number of splits, such as the sampling bias, size of the test-set and dimensionality of the HP space. For some of our benchmarks, the best results are reached with fewer splits, as shown in Figure 6 where improvements in both metrics are made until Split 2 and by splitting deeper, estimates get less accurate especially when sample sizes in sub-regions become very small. Thus, the number of splits is a (useful and flexible) control parameter in our method which can be determined within a human-in-the-loop approach (view plots after each split and stop when results are satisfying) or by defining a quantitative measure (e.g. based on a threshold for confidence improvement).

## Appendix D. Code

All code related to this paper is made available via an anonymous repository[5]. All methods are implemented within the folder `R`, and all code used to perform the experiments are provided in `benchmarks`. All analyses shown in this paper in form of tables or figures can be reproduced via running the notebooks in `analysis`.

---

5. `https://anonymous.4open.science/repository/a71006d4-e8df-475e-9848-03786f00bf99/README.md`

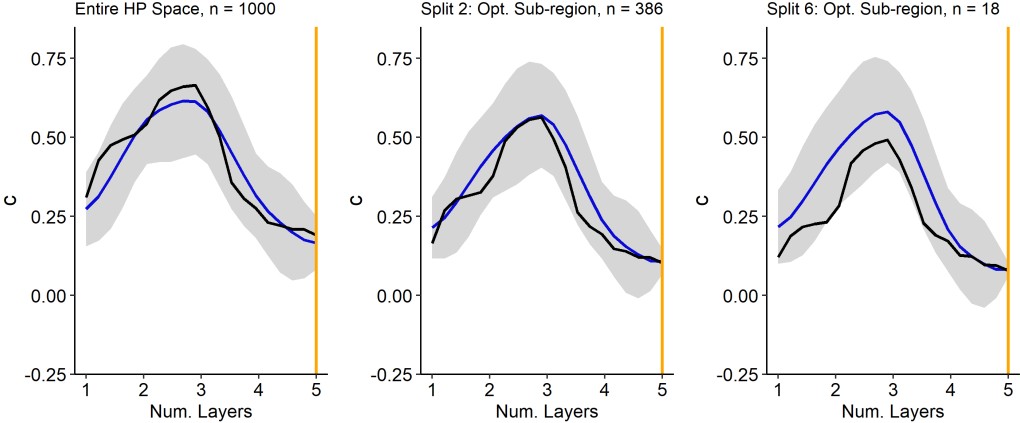

Figure 6: Estimated PDP of GP (blue) and true PDP estimate (black). The relative improvements after 2 (6) splits are $\delta$ MC = 5% (0%) and $\delta$ NLL = 5% ($-28\%$).

