# OpenReview forum: "Towards Explaining Hyperparameter Optimization via Partial Dependence Plots"
_ICML.cc/2021/Workshop/AutoML — AutoML@ICML2021 Poster_

### Official Review · Reviewer_HiSc · 2021-06-11
**A small but concrete contribution for boosting trust on hyperparameter auto-optimization**

**Rating:** 8
**Confidence:** 4

**Review:**

The paper attempts to provide and enrich explanations on how values of hyperparameters affect the performance of a machine learning model, which can help boost confidence on automated methods for searching for the optimal hyperparameter values.

The method is based on a well established method for explaining numerical prediction models, the partial dependence plot.  The numerical prediction model in concern is a surrogate model used with a Bayesian optimization procedure, which approximates the black-box machine learning model whose hyperparameters are to be optimized.   Knowing that the Bayesian optimization procedure tends to focus on promising areas, hence implying a sampling bias, this work proposes to use the posterior probability estimates from the Bayesian model to add error bars to a partial dependence plot that is used to explain effects of changes in a hyperparameter on the model performance.   This is expected to give users quantitative understanding of the uncertainty of the optimization, and enable the user to exercise active control during the automated search process.

Furthermore, the work proposes to split the hyperparameter value space into certain and uncertain regions based on this quantification and the CART algorithm, so as to provide finer-grain, localized understanding and control.

Experimental evidences are shown using a surrogate benchmark dataset that includes 2000 sampled configurations of
different learning architectures and hyperparameters evaluated on 35 OpenML datasets, with random forests as the surrogate models.   The results show improvements in two notions of confidence and the objective function in the subregion that contains the chosen optima of the hyperparameter estimates.

The work is a small but concrete addition to a technique for automating hyperparemeter search in machine learning.   It appears to be one intermediate step in a larger pursuit of an end-to-end, explainable autoML strategy. The paper is well written, though, it could be more helpful if the proposal is explained with a simpler example  (simple blackbox model with visualizable input, output, and few hyperparemeters for search) that is followed through.  As is, the results are claimed only in aggregate statistics and a few spotty values, which does not convey a easily understandable explanation of how THIS proposal works.

---

### Official Review · Reviewer_8ga2 · 2021-06-15
**A valuable contribution towards the understanding of the AutoML systems**

**Rating:** 7
**Confidence:** 3

**Review:**

In this paper, the authors propose an improvement in the Partial Dependence Plots Analysis to explain beyond the hyperparameter optimization (HPO) itself. They propose a tree-based partitioning algorithm to identify the more reliable sub-regions. Experiments were performed with Auto-PyTorch and LCBench data, with configuration samples over different datasets. Results are quite interesting, promising.

The paper is well-written and justified. The message is clear most of the time. The authors provide a valuable contribution to the AutoML research area aiming to "clarify" such black-box processes as HPO.

However, in my humble opinion, and feel free to agree or disagree, it focuses mainly on formalizing the problem (improving the PDPs). I mean, most of the details regarding the empirical experiments are in the Appendices. With a balance between them (formalization and experiments), the paper may reach a wider audience (AutoML experts and non-experts).

Below there are some other questions and suggestions.

1) What are the ICE curves often discussed during the paper? What does the acronym mean, and how is the concept important for further explanations? It would help to introduce them at some point, please.

2) In the related works section, the authors can provide more information about the related papers. For example, how Cafri & Bayley explored the bootstrap trees to handle the uncertainty measurement? The same could be expanded when describing different studies. I know we always work with few-page papers, but

3) Would it be possible to apply the improved PDPs in optimization methods different from BO? I do not get any comment about this and if it could be explored to explain different optimization process with population-based or racing methods.

4) Figures, in general, explain better the information they summarize. What are the black dots in Figure 1? How do they relate to the colored lines? Each would be a sample? Maybe give more detail when explaining those curves with different uncertainty levels.

5) GAMLSS - write the acronym in full when it appears for the first time

6) What are the main reasons to select Auto-PyTorch as a case study? Why not a different AutoML system? Auto-WEKA, Auto-skLearn, TPOT?

7) "As shown in Table 1, the MC of the PDPs is on average reduced by 30 to 52, depending on the hyperparameter".  Which one did reduce the MC: the global PDP or the splitted-PDP?

8) "NLL even improves while the MC decreases." Not totally clear here. What is expected according to the authors' hypothesis? Using splitted-PDP will increase OC and NLL while reducing MC? Is that the point?

9) How can non-expert users interpret such findings when evaluating their AutoML jobs? For example, in Figure 2, which situation is preferable for them?

---

### Decision · Program_Chairs · 2021-06-21

Accept (Poster)